# GBaTSv2: A revised synthesis of the likely basal thermal state of the Greenland Ice Sheet

Joseph A. MacGregor[1], Winnie Chu[2], William T. Colgan[3], Mark A. Fahnestock[4], Denis Felikson[1,5], Nanna B. Karlsson[3], Sophie M.J. Nowicki[6], Michael Studinger[1]

[1]Cryospheric Sciences Laboratory, NASA Goddard Space Flight Center, Greenbelt, Maryland, United States of America
[2]Department of Earth and Atmospheric Sciences, Georgia Institute of Technology, Atlanta, Georgia, United States of America
[3]Geological Survey of Denmark and Greenland, Copenhagen, Denmark
[4]Geophysical Institute, University of Alaska Fairbanks, Fairbanks, Alaska, United States of America
[5]Universities Space Research Association, Columbia, Maryland, United States of America
[6]Department of Geology, University at Buffalo, Buffalo, New York, United States of America

*Correspondence to*: Joseph A. MacGregor (joseph.a.macgregor@nasa.gov)

**Abstract.** The basal thermal state (frozen or thawed) of the Greenland Ice Sheet is under-constrained due to few direct measurements, yet knowledge of this state is becoming increasingly important to interpret modern changes in ice flow. The first synthesis of this state relied on inferences from widespread airborne and satellite observations and numerical models, for which most of the underlying datasets have since been updated. Further, new and independent constraints on the basal thermal state have been developed from analysis of basal and englacial reflections observed by airborne radar sounding. Here we synthesize constraints on the Greenland Ice Sheet's basal thermal state from boreholes, thermomechanical ice-flow models that participated in the Ice Sheet Model Intercomparison Project for CMIP6 (ISMIP6), BedMachine v4 bed topography, Making Earth Science Data Records for Use in Research Environments (MEaSUREs) multi-year surface velocity mosaic v1, and multiple inferences of a thawed bed from airborne radar sounding. Most constraints can only identify where the bed is likely thawed rather than where it is frozen. This revised synthesis of the Greenland likely Basal Thermal State version 2 (GBaTSv2) indicates that 33% of the ice sheet's bed is likely thawed, 40% is likely frozen, and the remainder (28%) is too uncertain to specify. The spatial pattern of GBaTSv2 is broadly similar to the previous synthesis, including a scalloped frozen core and thawed outlet-glacier systems. Although the likely basal thermal state of nearly half (46%) of the ice sheet changed designation, the assigned state changed from likely frozen to likely thawed (or vice versa) for less than 6% of the ice sheet. This revised synthesis suggests that more of northern Greenland is likely thawed at its bed, and conversely that more of southern Greenland is likely frozen, both of which influence interpretation of the ice sheet's present subglacial hydrology and models of its future evolution. The GBaTSv2 dataset, including both code that performed the analysis and the resulting raster products, is freely available at https://doi.org/10.5281/zenodo.5714527.

# 1    Introduction

The basal interface of an ice sheet is a fundamental control upon its flow and response to external forcings. As such, the ice-sheet bed is a perennial focus of much glaciological fieldwork and modeling studies, especially its lithology, hydrology and morphology, along with spatiotemporal variability in those properties (e.g., Cuffey and Paterson, 2010). However, the relevance of most basal properties to modulating ice flow is often predicated on the basal temperature being at or very near the pressure-melting point, i.e., a "thawed" basal thermal state. In other words, the bed is only as significant to ice flow as its temperature permits. If the bed is frozen and does not permit significant basal motion or subglacial water flow, then neither its roughness or rheology are likely to significantly influence ice flow at sub-centennial time scales. Resolving an ice sheet's basal thermal state is thus a prerequisite to interpretation of large-scale investigations of most other basal properties and processes.

For the Greenland Ice Sheet (GrIS), MacGregor et al. (2016) (hereafter M16) generated the first synthesis of its likely basal thermal state (GBaTSv1) from a combination of three-dimensional (3-D) thermomechanical ice-flow models, radiostratigraphy modeling, and surface-velocity and surface-texture analyses. The value of this synthesis lay not in its (in)certitude, but in its reduction of the substantial challenge of constraining basal temperature across an entire ice sheet to a simpler ternary determination of the likely basal thermal state of the GrIS. GBaTSv1 has served as a baseline for more sophisticated and localized interpretations of basal properties (e.g., Jordan et al., 2018; Chu et al., 2018; Oswald et al., 2018 Bowling et al., 2019), context for other observations of the GrIS (e.g., Bons et al., 2018; Leysinger-Vieli et al., 2018; MacGregor et al., 2020; Maier et al., 2021; Karlsson et al., 2021), and as a conceptual framework for investigations of former ice sheets (e.g., Menzies et al., 2020).

Since the generation of GBaTSv1, most of the key datasets that underlie its synthesis have been updated, and some of its inputs warrant reconsideration following subsequent independent analyses. In terms of direct observations of the GrIS interior, a new borehole (EastGRIP) is being drilled to the bed within the Northeast Greenland Ice Stream (NEGIS), additional observations of the base of the penultimate deep interior borehole (NEEM) have come to light. Additional older boreholes have been identified, newer boreholes have been drilled, and new subglacial lakes haven been identified. GBaTSv1 used 3-D thermomechanical model outputs from the Sea-Level Response to Ice Sheet Evolution (SeaRISE) project, which are now effectively superseded by those from the Ice Sheet Model Intercomparison Project for the Coupled Model Intercomparison Project 6 (ISMIP6). An improved synthesis of GrIS thickness has been generated (BedMachine v4; Morlighem et al., 2017, 2021) relative to that used previously (BedMachine v1; Morlighem et al., 2014). A new, complete long-term surface-velocity field for the GrIS is available from the NASA Making Earth Science Data Records for Use in Research Environments (MEaSUREs) program (Joughin et al., 2016, 2017). Multiple new studies of airborne radar-sounding data have since been conducted to identify either basal water or deep englacial structures potentially related to a thawed bed (Panton and Karlsson, 2015; Oswald et al., 2018; Jordan et al., 2018; Leysinger-Vieli et al., 2018; Bowling et al., 2019). Finally, recent investigations of basal roughness beneath the GrIS and the transmission of that roughness to the surface warrant a reevaluation of whether surface texture is a reliable indicator of non-negligible basal motion and hence a thawed bed (Ng et al., 2018; Ignéczi et al.,

2018; Cooper et al., 2019a, 2019b). Here we generate a new synthesis of the likely basal thermal state of the GrIS (GBaTSv2) using these new and updated datasets and refined methods. We then consider its differences relative to GBaTSv1 and its implications for interpretation for the present and future flow of the GrIS.

## 2 Data and methods

### 2.1 Direct observations of basal thermal state

As for M16, we consider "direct" observations of the GrIS basal thermal state (and that of Greenland's peripheral ice masses) to include both observations and inferences from deep boreholes, along with unambiguous evidence for subglacial lakes. Except for NEEM (discussed below), we use the same borehole and subglacial lake observations included in M16 (their Table 1). We further include basal thermal state information from 14 additional boreholes (EastGRIP, discussed below, and 13 other marginal boreholes) and two additional active subglacial lakes reported by Bowling et al. (2019) (Table 1). The 54 radar-identified subglacial lakes reported by Bowling et al. (2019) are considered in Sect. 2.4 and included as part of a broader synthesis of radar-based inferences of a thawed basal thermal state.

**Table 1:** Additional direct observations of basal thermal state from deep boreholes and subglacial lake detections.

| Site | Latitude (ºN) | Longitude (ºW) | Ice thickness (m) | Basal temperature (observed/corrected [1], ºC) | Reference |
|------|------|------|------|------|------|
| *Boreholes* | | | | | |
| EastGRIP | 75.63 | 35.99 | 2668 | T | Zeising and Humbert (2021) |
| Isua 10 | 65.2093 | 49.7500 | 97 | –2.3 / –2.2 | Colebeck and Gow (1979) |
| Isua 11 | 65.2072 | 49.7510 | 120 | –1.0 / 0.9 | *ibid.* |
| Isua 12 | 65.2039 | 49.7530 | 97 | –1.3 / –1.2 | *ibid.* |
| Isua 13 | 65.2069 | 49.7456 | 265 | T | *ibid.* |
| Isua 14 | 65.2058 | 49.7443 | 299 | T | *ibid.* |
| TD1 | 69.45 | 50.13 | 300 | T | Thomsen et al. (1991) |
| TD2 | 69.45 | 50.10 | 470 | T | *ibid.* |
| TD3 | 69.48 | 50.00 | 350 | T | *ibid.* |
| TD4 | 69.53 | 49.68 | >600 | T | *ibid.* |
| TD5 | 69.57 | 49.30 | >600 | T | *ibid.* |
| Store S30 | 70.520 | 49.920 | 611 | –0.5 / 0 | Doyle et al. (2018) |
| Store R30 | 70.57 | 50.09 | 1043 | –0.8 / 0 | Law et al. (2021) |
| Hans Tausen Hare | 82.84 | 36.67 | 289 | –1.7 / –1.4 | Reeh et al. (2001) |
| *Subglacial lakes* | | | | | |
| Sioqqap Sermia 1 | 63.54 | 48.45 | 722 | T | Bowling et al. (2019) |
| Sioqqap Sermia 2 | 63.26 | 48.21 | 1277 | T | *ibid.* |

[1] As in M16, "corrected" means adjusted for pressure melting using the local ice thickness. "T" means that the basal temperature was not measured directly but that a thawed bed can be confidently inferred.

Since GBaTSv1, two key borehole observations of the GrIS interior have arisen from its two most recent deep boreholes: NEEM and EastGRIP (Fig. 1). The ice thickness at NEEM is ~2538 m, indicating a pressure-melting point of –2.2ºC (assuming a decrease of $8.7 \times 10^{-4}$ K m$^{-1}$; Cuffey and Paterson 2010, p. 406). Drilling was completed in 2012 and repeat logging of

borehole temperature after the 2011 profile reported by MacGregor et al. (2015a) confirms a basal temperature of ~ –3.5ºC, inferred from the deepest englacial thermistor. However, subsequent logging directly at the base measured a higher temperature of –2.4ºC, presumed to be due to the presence of subglacial water (Colgan et al., 2022). Combined with the recovery of several meters of refrozen, debris-rich ice from the bottom of the NEEM core (D. Dahl-Jensen, pers. comm., 2021), these observations indicate that the base of the NEEM ice core is thawed, rather than frozen as previously estimated by M16. This change in identified basal thermal state at NEEM also implies that the temperature threshold for assuming the bed is thawed in 3-D thermomechanical models should be lower than previously assumed by M16 (Sect. 2.7).

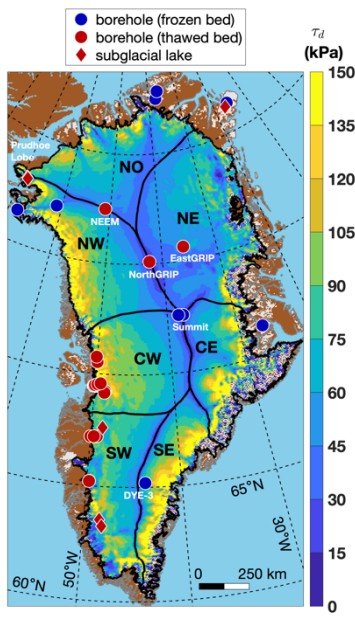

**Figure 1:** Reference map for GrIS study area with driving stress overlain (Sect. 2.5) and locations discussed in the text labeled. Ice-drainage basins are outlined and labeled following Mouginot et al. (2019). The reported basal thermal state of boreholes follows M16 and Table 1, except for NEEM and EastGRIP (Sect. 2.1). The four known subglacial lakes included in M16 are shown, along with two additional active subglacial lakes identified by Bowling et al. (2019).

While the basal thermal state at EastGRIP has not yet been directly measured by borehole thermometry, ice-core drilling there is underway (80% of ice thickness as of September 2021). Preliminary interpretation of the core's depth–age relation indicates that the bed there is thawed, an approach that has correctly predicted the basal thermal state in the past (Dahl-Jensen et al., 2003). Recent phase-sensitive radar measurements also indicate that basal melting is occurring there (Zeising and Humbert, 2021), so we assume that EastGRIP is indeed thawed for this study (Table 1).

## 2.2    3-D thermomechanical modeling of basal temperature

Since GBaTSv1, it remains the case that only 3-D thermomechanical numerical models can estimate basal temperatures beneath the entire ice sheet. To do so requires explicitly solving coupled mass-, momentum- and energy-conservation equations using imperfectly known initial conditions, boundary conditions and constitutive relations. This challenge is met by multiple families of ice-sheet models, of which the most recent and suitable ensemble is ISMIP6 (Goelzer et al., 2020; Nowicki et al., 2020). For the GrIS, ISMIP6 constitutes a 21-member ensemble of nine different model families. For the purposes of generating GBaTSv2, such an ensemble is strongly preferred over multiple instances of a single model, as it permits evaluation of a wider range of models with varying ice-flow parameterizations and numerical schemes, whose outputs were homogenized prior to the ensemble analysis. Several of the models used in the SeaRISE ensemble are no longer developed actively (Nowicki et al., 2013), further motivating a transition to the ISMIP6 ensemble. While the choice of ensemble is new, similar trade-offs exist as for the previous ensemble, i.e., variability in initialization and data-assimilation strategies and prescribed boundary conditions (e.g., Table A1 of Goelzer et al., 2020). As for GBaTSv1, we explicitly accept and welcome this diversity of model implementations, and here simply evaluate their agreement with one another.

Table 2 lists the ten model instances (hereafter simply "models") from the ISMIP6 ensemble that we consider for GBaTSv2 and our rationale for their selection. Most groups participating in ISMIP6 submitted multiple models with different spatial resolutions or stress-balance approximations; however, basal temperature outputs were not available for all model instances. We selected a single model from each participating group that we assessed to be the most physically complete (e.g., higher-order stress balance) or had the finest spatial resolution. Following M16, for each ISMIP6 model we only consider the modeled basal temperature on grounded ice ("litempbotgr" in ISMIP6 nomenclature) at the end of their 86-yr control runs ("ctrl_proj" in ISMIP6 nomenclature), which are intended to simulate the unforced state of the GrIS at the end of 2100 CE. We assume that this temperature accommodates further thermodynamic relaxation following spin-up, without additional external forcing. Modeled basal temperature ($T_b$) is corrected upward for pressure melting ($T'_b$) using the contemporaneous modeled ice thickness ($H$) and assuming a melting-point decrease of $8.7 \times 10^{-4}$ K m$^{-1}$ (Fig. 2).

**Table 2:** ISMIP6 ensemble model instances considered for GBaTSv2.

| Institution [1] | Model instance | Rationale for selection |
|---|---|---|
| AWI | ISSM3 [2] | Finest resolution model interpolated from a paleoclimatically forced thermal spin-up |
| JPL | ISSMPALEO | Spin-up across the entire Last Glacial Period |
| UCIJPL | ISSM1 | No drainage-specific sub-modeling |
| UAF | PISM2 [2] | Open forcing framework instead of retreat parameterization |
| VUW | PISM | Only model submitted from this institution |
| MUN | GSM2601 [2] | More sophisticated representation of basal sliding |
| VUB | GISMHOMv1 | Higher-order stress-balance approximation |
| ILTS_PIK | SICOPOLIS3 | Paleoclimatic spin-up with higher-order stress-balance approximation |
| LSCE | GRISLI2 [2] | N/A |
| NCAR | CISM | Only instance submitted from this institution |

[1] AWI: Alfred-Wegener-Institut; JPL: Jet Propulsion Laboratory; UCIJPL: University of California, Irvine and Jet Propulsion Laboratory; UAF: University of Alaska, Fairbanks; VUW: Victoria University of Wellington; MUN: Memorial University of Newfoundland; VUB:

Vrije Universiteit Brussel; ILTS_PIK: Institute of Low Temperature Science, Hokkaido University and JP/Potsdam Institute for Climate Impact Research; LSCE: Laboratoire des Sciences du Climat et de l'Environnement; NCAR: National Center for Atmospheric Research.

2 The difference in $T'_b$ between the selected model and others from this institution is small.

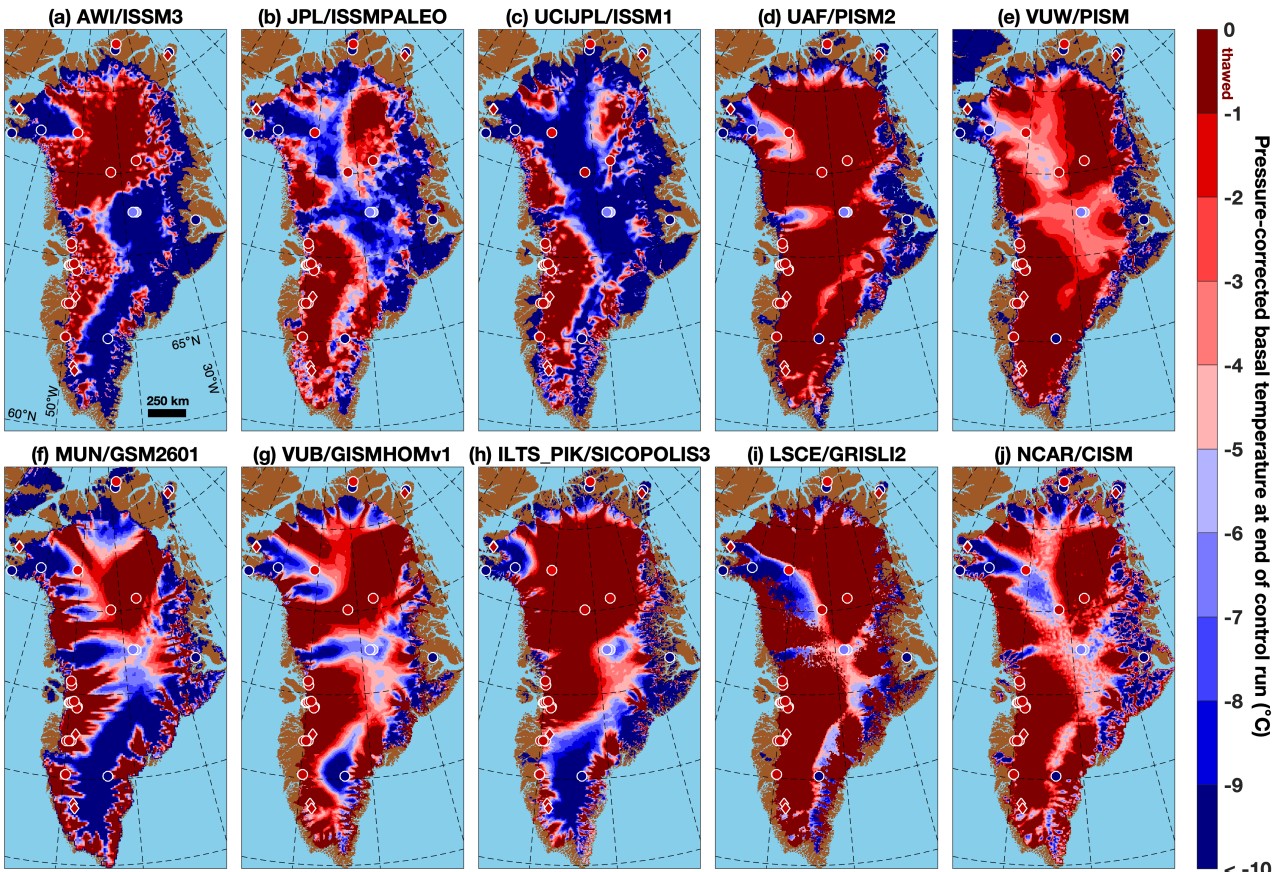

Figure 2: Modeled basal temperature ($T'_b$) across the GrIS at the end of ten IMSIP6 control-run experiments, corrected for pressure-melting using each instance's ice-thickness field. The highest temperature range shown (darkest red) represents the range of basal temperatures we
consider thawed for our "standard" temperature threshold. Symbology follows Fig. 1, except that the color for borehole symbols instead follows the color scale for their reported or apparent values of $T'_b$.

As in M16, the agreement in basal temperature between the ten models selected from the ISMIP6 ensemble is then combined for subsequent inclusion in a multi-method synthesis (Fig. 3). This pattern is qualitatively similar to that of M16 (their Fig. 4) but shows generally greater model agreement overall and more tightly defined thawed regions for some northern,
eastern and southeastern outlet glaciers. Part of the key difference between this study and M16 lies in the selection of the temperature thresholds for identifying a thawed bed. Our reinterpretation of the NEEM bed as thawed, despite a basal temperature >1 K below the pressure-melting point, suggests that M16's temperature thresholds were too conservative, i.e., they erred on the side of a frozen bed identification. We thus select –1ºC below the pressure-melting point as the standard

temperature threshold for identifying a thawed bed (M16 used –0.05ºC), and increase the range considered for the cold- (–0.5ºC) and warm-bias (–1.5ºC) thresholds. This adjustment acknowledges greater uncertainty in basal thermal state from directly measured borehole temperatures (e.g., Sect. 2.1), which implies greater ambiguity in interpretation of modeled basal temperatures.

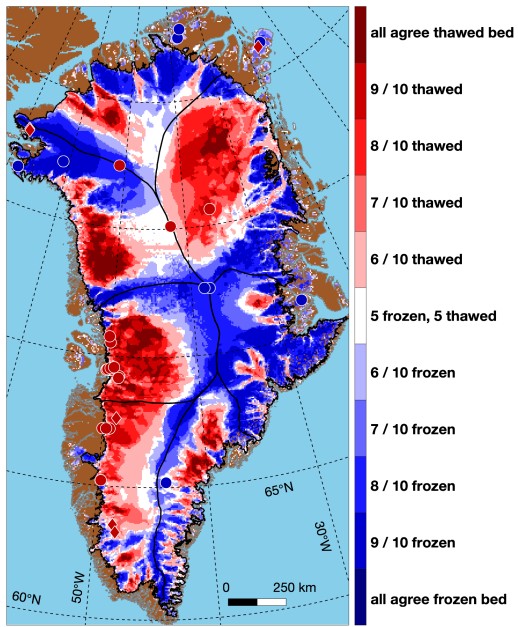

**Figure 3:** Agreement in modeled basal thermal state between the ten selected ISMIP6 control-run experiments (Fig. 2), assuming that the bed is thawed where $T'_{bed} \geq -1ºC$ and frozen where $T'_{bed} < 1ºC$.

## 2.3 Basal melting from radiostratigraphy

M16 used one-dimensional (1-D) steady-state modeling of radar-observed Holocene (9–0 ka) depth–age relations to constrain the multi-millennial scale pattern of ice flow across a broad swath of the GrIS interior (69% by area), which can indirectly constrain its basal thermal state. The two primary models used to interpret these depth–age relations, "Dansgaard–Johnsen" (Dansgaard and Johnsen, 1969) and "Nye + melt" (Fahnestock et al., 2001), are both two-parameter representations of vertical strain with differing underlying assumptions about local ice flow. However, for the purposes of constraining basal thermal state, they are fundamentally related. As shown by Fahnestock et al. (2001) and M16, a best-fit Dansgaard–Johnsen model that infers a negative basal shear-layer thickness ($h < 0$) is qualitatively comparable to a best-fit Nye + melt model that infers a positive basal melt rate ($\dot{m} > 0$), whereas vice versa implies non-negligible basal freeze-on ($\dot{m} < 0$). M16 recast the basal shear-layer thickness of the Dansgaard–Johnsen model as a geometric shape factor $\phi$ for the horizontal ice flow of the bulk column. This interpretation offered the potential to constrain not only where the bed is thawed ($\phi > 1$) but also where the bed is frozen, because the natural lower limit for $\phi$ should be $(n + 1)/(n + 2) \approx 0.8$ (Cuffey and Paterson, 2010, p. 310), where $n$ is the

flow-law exponent and assumed to be 3 (Sect. 2.5). However, a surprisingly large fraction of the interpretable area (57%) displayed $\phi$ values below this limit, calling into question the assumptions underlying that interpretation of $\phi$.

For GBaTSv2, we retreat from the possible interpretation of a frozen basal thermal state from radiostratigraphy and instead focus only where these data clearly indicate basal melting and hence a thawed bed. This simplifies interpretation of Holocene radiostratigraphy to using the Nye + melt model only and provides a straightforward significance cutoff for interpreting a thawed bed, i.e., where $\dot{m} > 0$ cm yr$^{-1}$, which we conservatively increase to where $\dot{m} \geq 1$ cm yr$^{-1}$ (regions with red coloring in Fig. 4). Conversely, apparent $\dot{m}$ values inferred from radiostratigraphy indicate large regions where $\dot{m} < 0$ (Fig. 4). However, as explained by M16, those values should be interpreted primarily as due to a limitation in interpretation of the Nye + melt model in regions where there is non-negligible basal shear, rather than an indicator of widespread, rapid basal freeze-on. This caution is further supported by the independent modeling study of Dow et al. (2018), which indicates that the mean basal freeze-on rate across the GrIS is $< 0.02$ cm yr$^{-1}$.

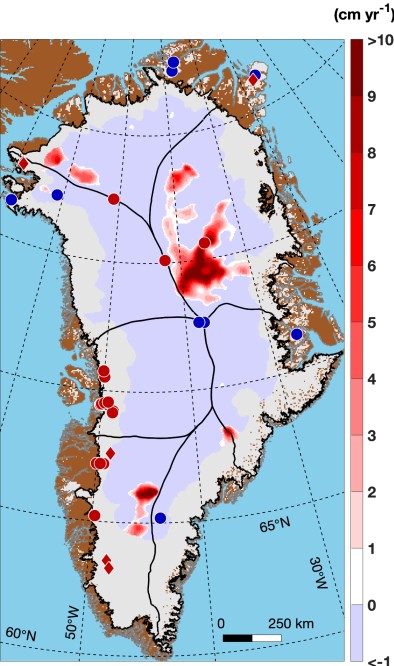

**Figure 4:** Gridded apparent basal melt rate ($\dot{m}$) from 1-D steady state modeling of Holocene (9–0 ka) radiostratigraphy using the Nye + melt model. An ice-thickness-dependent triangular filter has been applied to this dataset (Sect. 2.5).

While the focus of the interpretation has changed, the underlying dataset has not. Here we use the same GrIS dated radiostratigraphy dataset from MacGregor et al. (2015b) considered in M16, because no revision to the GrIS radiostratigraphy dataset yet exists. Uncertainty in $\dot{m}$ is reflected by the range between its lower- and upper-bound values ($\dot{m}_{min}$ and $\dot{m}_{max}$),

which are determined from the 95% confidence bounds for this model parameter in the Nye + melt model and are the same as for M16.

## 2.4    Basal water from airborne radar sounding

Since M16, multiple studies have mapped the apparent presence of basal water across the GrIS from analysis of airborne radar-sounding data, including investigations of bed reflectivity (Jordan et al., 2018; Oswald et al., 2018; Bowling et al., 2019), the morphology of the ice–bed reflection (Bowling et al., 2019), and indirectly via the identification of disrupted basal ice (Panton and Karlsson, 2015; Leysinger-Vieli et al., 2018). Airborne survey coverage is often sparse in the GrIS interior, where large gaps persist that can be tens of kilometers wide; further, at finer scales ($< \sim 50$–100 km) there can be notable differences in the inferred location of basal water between individual studies. However, at the scale of the whole ice sheet, the ensemble of the above analyses shows reasonable agreement and can therefore be credibly synthesized to interpret where airborne radar sounding has found evidence of basal water and hence a thawed bed. We merge four of these basal water datasets into a single mask of the likelihood of the presence of basal water ($M_{bw}$; Fig. 5). Where nuanced results were reported, indicating various degrees of confidence in the individual datasets by the study authors, we attempt to preserve that nuance when merging them. We attempted to acquire the gridded basal water estimate of Oswald et al. (2018) (their Fig. 13) but were unsuccessful, so it is not included in our synthesis.

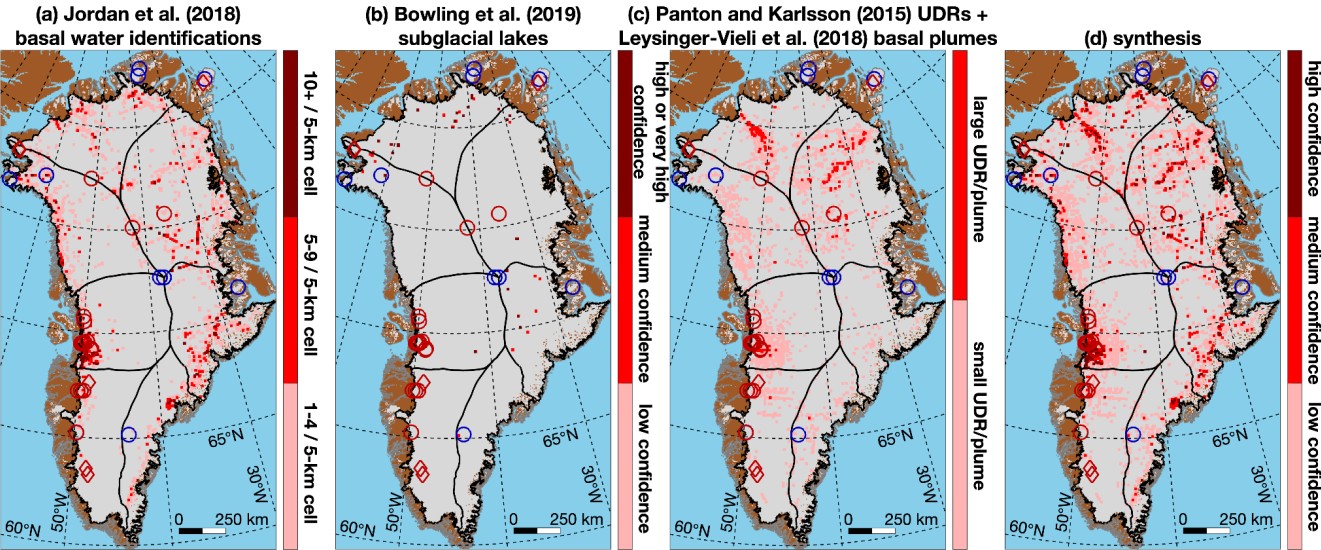

**Figure 5:** (a) Number of basal water identifications (Jordan et al., 2017) per 5-km grid cell. (b) Subglacial lakes identified by Bowling et al. (2019). (c) UDRs/basal plumes identified by either Panton and Karlsson (2015) or Leysinger-Vieli et al. (2018). (d) Merged inferences of presence of basal water from analysis of bed reflections or deep radiostratigraphy, respectively, in NASA airborne radar-sounding data

($M_{bw}$). A value of 1–4 for $M_{bw}$ indicates low confidence, 5–9 indicates medium confidence, and >10 is assigned high confidence. Symbology follows Fig. 1, except that open symbols are used so that underlying inferences of basal water can be better shown.

We use the basal water identifications of Jordan et al. (2018) (their Fig. 6), which include along-track binary identifications of basal water from basal radar reflectivity analysis of Operation IceBridge (OIB) and pre-OIB NASA airborne radar-sounding surveys. We binned these identifications into a 5-km grid by the total number of identifications within the nearest grid cell (Fig. 5a). For larger subglacial water bodies, Bowling et al. (2019) synthesized evidence for subglacial lakes beneath the GrIS

using multiple well-established criteria to analyze ice–bed reflections in OIB and pre-OIB NASA radar-sounding data. They assigned four possible confidence levels to their identifications: "low", "medium", "high" and "very high" (their Fig. 3). To render these confidence levels compatible with the other datasets, we re-assigned these confidence levels to values of 1, 5, 9 and 10, respectively. We then add those values to the nearest 5-km grid cell (Fig. 5b). In this manner, the contributions to the synthesis of basal water estimates are roughly equalized.

Finally, we include two separate maps of disrupted basal ice by Panton and Karlsson (2015) and Leysinger-Vieli et al. (2018) (Fig. 5c). Panton and Karlsson (2015) automatically identified units of disrupted radiostratigraphy (UDRs, which were invariably most disrupted near the bed) across the GrIS from 1999–2014 NASA pre-OIB and OIB data, whereas Leysinger-Vieli et al. (2018) examined 2010–2014 OIB data across the northern GrIS only to detect both "small" and "large" basal plumes. Panton and Karlsson (2015) remained agnostic as to the origin of the detected UDRs, whereas Leysinger-Vieli et al.

(2018) further analyze the structure of their identified basal plumes and conclude that they are most likely initiated by basal freeze-on. While the significance of basal freeze-on is controversial (e.g., Dahl-Jensen et al., 2013; Bons et al., 2016; Dow et al., 2018), it remains possible that the genesis of these features could require locally sourced basal water and hence a thawed basal thermal state. In northern Greenland, the patterns of disrupted radiostratigraphy from Panton and Karlsson (2015) and Leysinger-Vieli et al. (2018) are qualitatively similar, so here we simply merge the maps from both studies. We assume that

the putative basal water source that initiated these features still exists, and we neglect any horizontal displacement of location of that source relative to their identified location (typically the apex). Following the nomenclature of Leysinger-Vieli et al. (2018), we bin UDR/plume identifications to the nearest 5-km grid cell and add to them a value of either 1 (for small plumes) or 5 (large). For the UDR identifications of Panton and Karlsson (2015), we ignore regions where ice thickness is less than 1 km, due to a lower signal-to-noise ratio there. We assume all UDRs represent small plumes, except where the ratio of their

height above the bed to the ice thickness exceeds ⅓ $H$, which is the threshold selected by Leysinger-Vieli et al. (2018) for identification of a large plume.

Given sparse survey coverage of the GrIS interior, we assume that evidence of local basal water implies that a broader region of the adjacent bed possesses similar evidence for basal water but is as-of-yet unsurveyed. For each summed bin, we assign all eight adjacent bins the same value, effectively assuming that value for any individual 5-km grid cell is valid within

235 a 15-km-square region centered on that grid cell (Fig. 5d). Similar strategies have been employed previously (e.g., Oswald et

al., 2018), although ours is somewhat more conservative in that the regional extrapolation of the basal water signal has a fixed and finite range.

## 2.5 Minimum basal slip ratio

For GBaTSv2, we follow the method introduced in M16 (with minor modifications) to model the ice column's maximum possible deformation speed ($u_d^T$) under the end-member assumption that the whole of the ice column is temperate and hence as soft as possible, without directly invoking additional rheological processes (e.g., crystal-orientation fabric or damage). Where the observed surface speed ($|\vec{u}_s|$) is greater than that hypothetical "speed limit", i.e., where the minimum basal slip ratio ($\gamma_{\min} = |\vec{u}_s|/u_d^T$) exceeds unity, this implies that non-negligible basal motion is occurring there and that the bed is likely
thawed.

Similar to M16, we calculate $u_d^T$ by assuming that the shallow-ice approximation is appropriate for large-scale estimates of the deformation speed as (Cuffey and Paterson, 2010, p. 310):

$$u_d^T = \frac{2E\bar{A}H}{n+1}(\rho_{\mathrm{ice}}gH\alpha)^n, \qquad (1)$$

where the bulk density of the ice column $\rho_{\mathrm{ice}}$ is 900 kg m$^{-3}$, the rate of acceleration due to gravity $g$ is 9.81 m s$^{-2}$, the rate
factor for temperate ice ($\bar{A}$) is 2.4 × 10$^{-24}$ Pa$^{-3}$ s$^{-1}$ and $E$ is the depth-averaged enhancement factor for the whole column, $H$ is ice thickness and $\alpha$ is surface slope in the ice-flow direction.

The three main modifications to M16 are the use of updated datasets, an adjusted filtering scheme, and a revision of the value of $E$ and uncertainty therein. First, we now use BedMachine v4 for $H$ (Morlighem et al., 2017, 2021) and the MEaSUREs multi-year velocity mosaic v1 for $\vec{u}_s$ (Joughin et al., 2016, 2017; Fig. 6a). Second, rather than using an exponentially decaying
thickness-dependent filter as in M16, we use a triangular-shaped filter of width $10H$, following the recommendation of McCormack et al. (2019) for filtering driving stress. We apply this filter to all input datasets except modeled $T'_b$ and $M_{bw}$ (Sect. 2.2 and 2.4). To determine $\alpha$, we first take the gradient of the Greenland Ice Mapping Project's surface-elevation model (GIMP; Howat et al., 2014), as for GBaTSv1. To reliably determine $\alpha$ in the direction of ice flow, we exponentially weight the surface-velocity azimuth toward that of the gradient of the GIMP surface elevation as $\exp(-|\vec{u}_s|/u_r)$, where $u_r$ is a
reference speed set to 100 m yr$^{-1}$. This weighting reduces the noise associated with less reliable surface-velocity azimuths in areas of slower ice flow, which includes most of the ice sheet and is also where the basal thermal state is most poorly constrained. Third, M16 assumed a value of unity for $E$ and that the relative uncertainty in the product of $E\bar{A}$ was 25%, but Cuffey and Paterson (2010, p. 74) indicate that this value of $E$ is too low for polar ice undergoing simple shear. Instead, here we treat $E = 1$ as a lower bound, assume a new default value of 2 and an upper bound of 4. Uncertainty in the value of $n$ is
considered separately below.

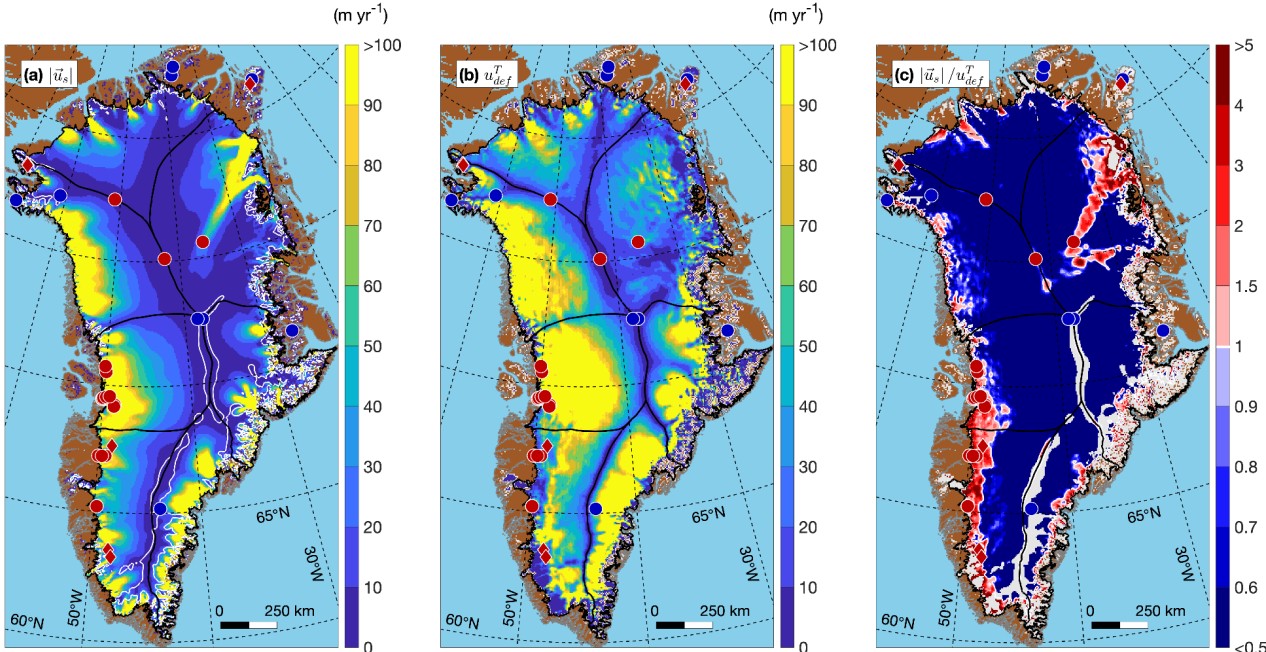

Figure 6: (a) Filtered MEaSUREs v1 observed surface speed ($|\vec{u}_s|$) across the GrIS. Regions where its relative uncertainty ($\tilde{u}_s/|\vec{u}_s|$) is $\geq$ 25% are outlined in white. (b) Modeled ice-deformation speed at the ice surface ($u_d^T$) assuming an entirely temperate ice column. (c) Ratio of observed surface speed to modeled deformation speed ($\gamma_{min}$). Regions where $\tilde{u}_s/|\vec{u}_s| \geq 25\%$ are masked out, and regions where $\gamma_{min} = 1$ are outlined in white.

Fig. 6 shows the filtered observed surface speed, modeled temperate-column deformation speed and the ratio of these two fields ($\gamma_{min}$). Where $\gamma_{min} > 1$, the ice column is inferred to exceed its speed limit due to deformation alone and that basal motion is occurring, implying a locally thawed bed. Assuming that regions where surface speed is poorly known limit the usefulness of this method, regions where the uncertainty in the surface speed ($\tilde{u}_s$) is $\geq$ 25% of $|\vec{u}_s|$ are not included in this method's assessment of the basal thermal state; these regions are located primarily along central and southern ice divides (Fig. 6a). Assumed and reported uncertainties in $u_d^T$ and $|\vec{u}_s|$, respectively, are used to calculate lower- and upper-bound values of $\gamma_{min}$ to then assess uncertainty in the basal thermal state agreement from this method. This analysis produces a substantially smaller region than M16 where $\gamma_{min} > 1$, principally because $u_d^T$ is roughly twice as large as was previously assumed, due to the change in assumed value of $E$.

Since M16 and GBaTSv1, Bons et al. (2018) inferred that $n \approx 4$ and $\bar{A} \approx 3.3 \times 10^{-29}$ Pa$^{-4}$ s$^{-1}$ for the northern GrIS based on an analysis of surface velocity, surface elevation and ice thickness within a bespoke reference area. If these alternative values are used in our $\gamma_{min}$ analysis, $u_d^T$ increases and $\gamma_{min}$ decreases, as Bons et al. (2018) concluded. We acknowledge that the value of the exponent in the flow "law" for large ice masses is uncertain across a range of flow regimes and timescales (e.g., Cuffey and Kavanaugh, 2011; Millstein et al., 2022), but we opt to continue using $n = 3$ because it permits conceptual continuity in our method for detecting where the ice exceeds its deformational "speed limit". The value of $\bar{A}$ inferred by Bons

et al. (2018) is a large-scale spatial average for the colder ice columns present in the northern GrIS (MacGregor et al., 2015a), and it cannot be simply disentangled from its associated $n$ value and then corrected using an Arrhenius relation to a presumed temperate value, which is necessary for our method of calculating $\gamma_{min}$. Further, the range of $E$ values that we now consider (1–4) produces substantially larger increases in $u_d^T$ with $n = 3$ than using the Bons et al. (2018) rheological parameters, so we consider this range more suitable for calculating $\gamma_{min}$.

## 2.6    Discontinued methods

As part of GBaTSv1, M16 mapped the onset of surface undulations across the GrIS from surface imagery, as they are suggestive – but not definitively indicative – of the onset of substantial basal motion and hence a thawed bed. Since M16, multiple additional studies have further explored both the nature of basal roughness beneath the GrIS (Cooper et al., 2019a), how that roughness is transmitted to the surface via basal motion (Ng et al., 2018; Ignéczi et al., 2018), and made independent observations of surface texture (e.g., Cooper et al., 2019b). When considered together with de Rydt et al. (2013), which formed part of the rationale for including this method in M16, we conclude that the onset of surface undulations can no longer be considered a reliable indicator of a thawed bed and we discontinue its use for GBaTSv2. Our rationale is explained further below.

Surface undulations due to ice flow over bedrock obstacles are expected to be more prominent where either basal roughness is more pronounced or the ratio of basal motion to the deformation speed is greater ($\gamma$, Sect. 2.5). Cooper et al. (2019a) found that basal roughness beneath the GrIS observed by airborne radar sounding at along-track scales of 200 m is typically greater within ~200 km of the ice margin than farther inland. Along-flow roughness is more likely to be efficiently transmitted to the surface than across-flow roughness (e.g., Ng et al., 2018), and the pattern of greater marginal roughness is less pronounced along-flow. Basal roughness at a 200-m horizontal scale is unlikely to generate significant surface undulations where the ice sheet is generally several times thicker than that (Ng et al. 2018). However, in northwestern Greenland, rougher marginal areas also have a higher degree of self-affinity, suggesting they are also rougher at larger horizontal scales (Jordan et al., 2017). Overall, these studies imply that *a priori* we should expect more surface undulations closer to the ice margin due to increasing basal roughness there, independent of any change in basal thermal state. Separately, Ng et al. (2018) and Ignéczi et al. (2018) refined modeling of bed-to-surface transmission and further emphasize the primary role of topography in generating modeled surface undulations that credibly reproduce observations, rather than those generated purely by a non-zero slip ratio. The value of outlining the onset of surface undulations for GBaTSv1 was predicated on the dominant role of the latter mechanism only. Finally, Cooper et al. (2019b) found evidence of the surface expression of englacial features (e.g., disrupted basal units) and subglacial channels oriented along-flow in northwestern Greenland. These expressions are clearly not surface undulations that might be diagnostic of a thawed basal thermal state yet can be easily confused for them. Similarly, Kjær et al. (2018) and MacGregor et al. (2019) demonstrated that the presence of subglacial impact craters can be discerned partly from their surface expressions, and these structures are not yet conclusively associated with a particular basal thermal state. In summary, we conclude that it is no longer clear whether outlining surface undulations can be considered a reliable method for

demarcating a low or negligible basal slip ratio ($\gamma \rightarrow 0$), for which no basal thermal state assignment can be made, from a high or non-negligible ratio ($\gamma \gg 0$) that clearly indicates a thawed bed.

## 2.7 Synthesizing basal thermal state estimates

We follow a similar methodology to M16 for generating GBaTSv2, with several minor adjustments. The thresholds for a positive identification of a particular basal thermal state are summarized in Table 3, including both the "standard" values and cold- and warm-bias values that consider uncertainty in each method; these are later used to assess the likelihood of a particular basal thermal state.

**Table 3:** Thresholds for inference of a particular basal thermal state.

| Method | Implies a frozen bed | Implies a thawed bed | Cold-bias threshold | Warm-bias threshold |
|---|---|---|---|---|
| 3-D thermomechanical model | $T'_b < -1\text{°C}$ [a] | $T'_b \geq -1\text{°C}$ [a] | $-0.5\text{°C}$ [a] | $-1.5\text{°C}$ [a] |
| Basal melt rate from radiostratigraphy | N/A | $\dot{m} \geq 1 \text{ cm yr}^{-1}$ | Same threshold, but evaluated against $\dot{m}_{min}$ | Same threshold, but evaluated against $\dot{m}_{max}$ |
| Basal water from airborne radar sounding [b] | N/A | $M_{bw} \geq 5$ | 10 | 1 |
| Minimum basal slip ratio | N/A | $\gamma_{min} \geq 1$ | Same threshold, but $\gamma_{min}$ estimated using $|\vec{u}_s| + \tilde{u}_s$ and $u_d^T(1 - \widehat{E\tilde{A}})$ | Same threshold, but $\gamma_{min}$ estimated using $|\vec{u}_s| - \tilde{u}_s$ and $u_d^T(1 + \widehat{E\tilde{A}})$ |

[a] Note these changes from GBaTSv1, which used –0.05°C, 0°C and 0.5°C as the standard, cold- and warm-bias thresholds, respectively.
[b] The standard (5), cold- (10) and warm-bias (1) thresholds are equivalent to the number of basal water identifications within each 5-km grid cell synthesized by $M_{bw}$.

We synthesize the four methods of constraining the likely basal thermal state of the GrIS by first assessing where they each produce a clear signal regarding this state (Table 3; Fig. 7a). We initialize a 5-km gridded ice-sheet mask $S$ to zero. For each method, if that method indicates a thawed bed, then +1 is added to $S$ (Fig. 7b). Conversely, if the method indicates a frozen bed, then –1 is added to $S$. For the ISMIP6 ensemble, the agreement is considered significant only where at least 7/10 models agree that the bed is either frozen or thawed (Fig. 3), a more conservative assessment from GBaTSv1, for which only a plurality (more than half) of the eight SeaRISE models had to agree to reach the same assessment. All 3-D models are weighted equally, as are each of the methods. This process of generating $S$ is repeated using the cold- and warm-bias thresholds to generate $S_{\text{cold}}$ and $S_{\text{warm}}$, respectively (Fig. 7c,d). For GBaTSv2, only one method can distinguish a frozen bed (3-D thermomechanical models), so the range of possible values for $S$ is less than for GBaTSv1.

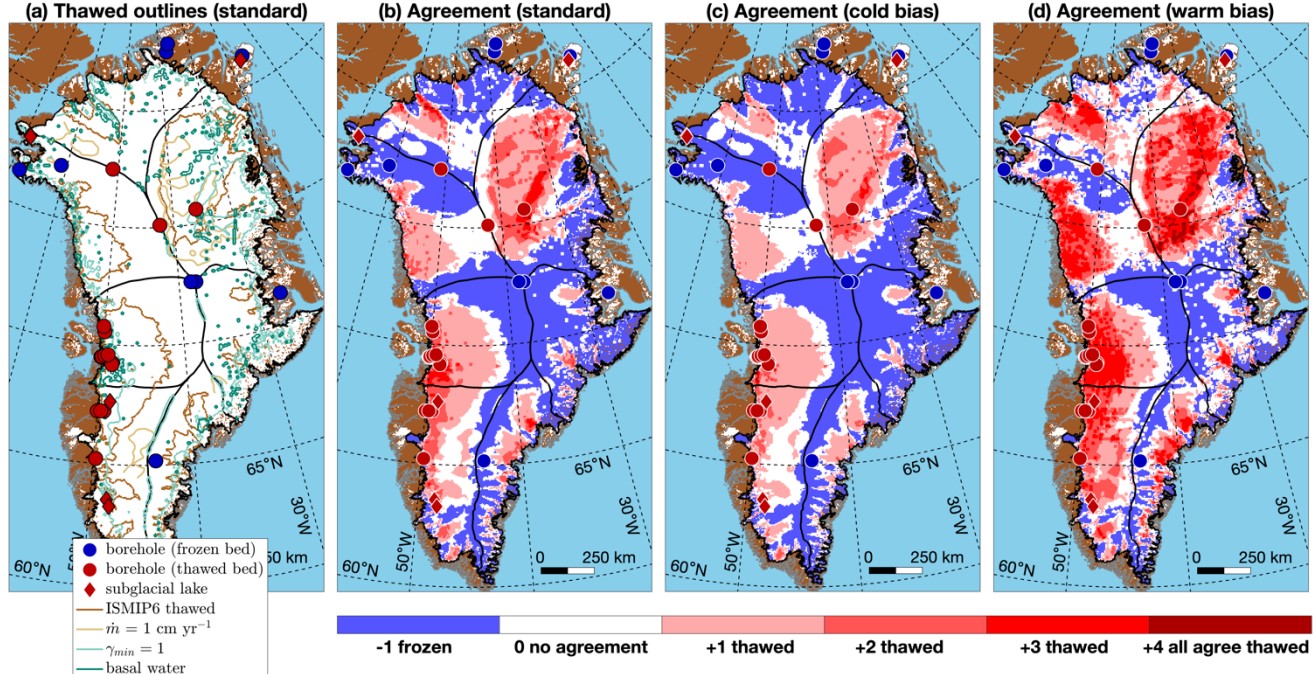

**Figure 7:** (a) Outlines of a thawed GrIS bed for the four methods considered in this study (Fig. 3, 4, 5d, 6c). For the ISMIP6 agreement, their outline denotes where at least 7/10 models agree that the bed is thawed. (b) Agreement between the four methods regarding the basal thermal state using standard thresholds ($S$; Table 3). (c, d) Cold- and warm-bias agreement ($S_\text{cold}$ and $S_\text{warm}$, respectively) determined using each method's confidence bounds or *ad hoc* uncertainty estimates. Because only one method constrains where the bed is frozen (thermomechanical models), but all four constrain where it is thawed, the range of possible values is –1 (frozen) to +4 (all thawed).

Based purely on $S$, $S_\text{cold}$ and $S_\text{warm}$, we generate the likely basal thermal state mask ($L$), which synthesizes their agreement and is the primary GBaTSv2 product. $L$ is initialized to zero (uncertain basal thermal state) and then assigned +1 for a likely thawed bed where both $S$ and $S_\text{warm}$ agree that the bed is thawed *and* $S_\text{cold}$ does *not* indicate that the bed is frozen. Similarly, $L$ is assigned –1 for a likely frozen bed where both $S$ and $S_\text{cold}$ agree that the bed is frozen and $S_\text{warm}$ does not contraindicate them. In other words, given our present understanding of the uncertainty of each method, we do not assign a likely basal thermal state if any of the three instances of $S$ contradicts the other two. We only consider the sign of $S$ and ignore its magnitude. Where $L$ contains small "holes" ($\leq 10$ grid cells, equivalent to $\leq 250$ km$^2$) in predominantly thawed regions, i.e., small regions with a different basal thermal state (uncertain or frozen), these are filled in as in M16 and assumed to be likely thawed. This process is then repeated for small holes in frozen regions, except those are assumed to be likely frozen. However, we do not repeat this process for holes in uncertain regions, as was done for GBaTSv1, because we infer that there is insufficient evidence to justify a particular assignment of basal thermal state there. These hole-filling procedures result in less than a 1% difference in the total area assigned to each basal thermal state.

## 3 Results

Fig. 8 shows version 2 of the likely basal thermal state of the GrIS (GBaTSv2), based on the four methods and their synthesis described in Sect. 2. At the scale of the whole ice sheet, this synthesis is qualitatively similar to GBaTSv1, but there are notable regional differences highlighted below and summarized by ice-drainage basin in Table 4. The most prominent differences are along the southern portion (≤ 68°N) of the central ice divide (more contiguous regions of likely frozen bed in GBaTSv2), west of the central ice divide that lies between Summit and NorthGRIP (less confidence in a frozen bed in GBaTSv2), and within the drainage basin that includes the NEGIS (NE; more contiguous regions of likely thawed bed northwest of NEGIS in GBaTSv2). Similarities between the two versions include large contiguous regions of likely thawed bed along the southwestern and northwestern coasts (up to Melville Bay), and within the NEGIS ice-drainage basin. The "scalloped frozen core" described by M16 is now potentially dissected between its southern and northern reaches, primarily due to the reduced agreement between 3-D thermomechanical models on the extent of the frozen-bedded region.

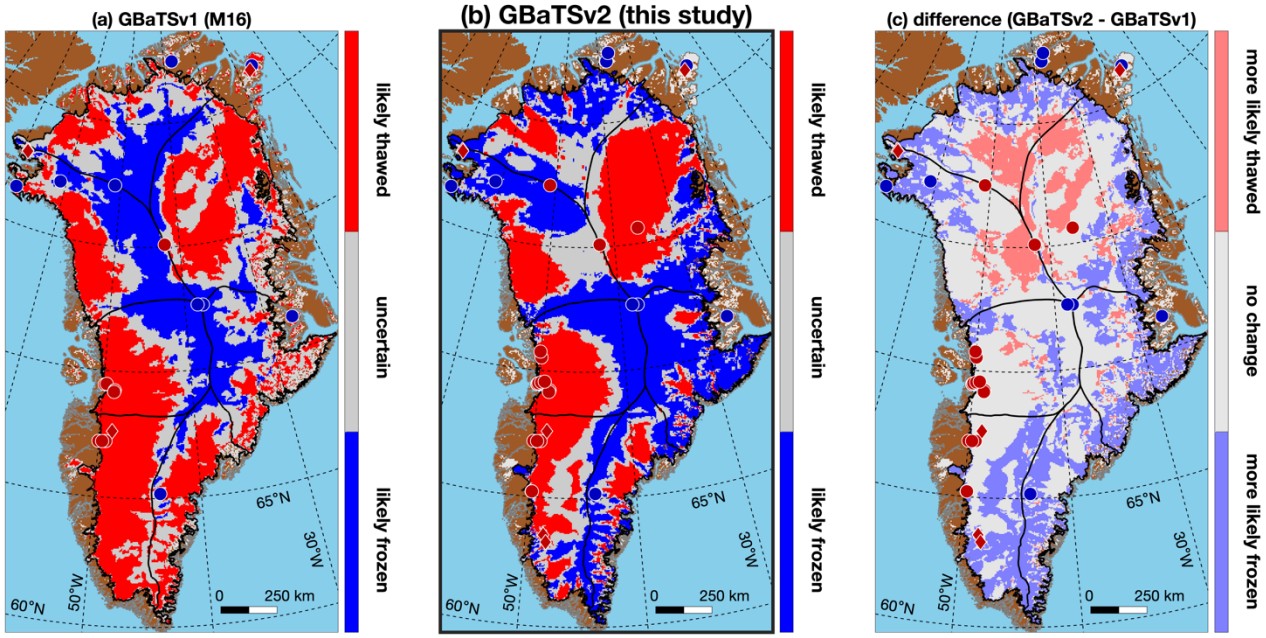

**Figure 8:** (a) Previous (GBaTSv1) and (b) new (GBaTSv2) likely basal thermal state of the GrIS (*L*), based on where the standard, cold- and warm-bias estimates of this state agree (Fig. 7b–d; Sect. 2.7). Note that (a) shows the borehole states and subglacial lakes as reported by M16. (c) Difference between GBaTSv2 and GBaTSv1.

**Table 4:** Areal percentage of GBaTSv2 likely basal thermal state by Mouginot et al. (2019) ice-drainage basin. [*]

| Ice-drainage basin | Likely frozen | Uncertain | Likely thawed |
|---|---|---|---|
| NW | 44 | 31 | 26 |
| NO | 43 | 42 | 14 |
| NE | 24 | 23 | 53 |

| | | | |
|---|---|---|---|
| CE | 74 | 19 | 8 |
| SE | 48 | 30 | 23 |
| SW | 22 | 36 | 42 |
| CW | 40 | 15 | 45 |
| GrIS total | 40 | 28 | 33 |

[*] Percentages are rounded to the nearest integer, so total areas shown here (sum of likely frozen, uncertain and likely thawed areas) may not equal 100%. While the likely basal thermal state of peripheral ice masses is shown in Fig. 8, this table reports values for the ice sheet only.

As compared to GBaTSv1 (M16), GBaTSv2 reports an uncertain basal thermal state at both NorthGRIP and DYE-3. However, it evinces greater uncertainty in the vicinity of NorthGRIP (especially farther west), but greater confidence that regions near DYE-3 are frozen. We interpret both changes as minor improvements in GBaTSv2 over GBaTSv1. However, we note two areas of concern in terms of GBaTSv2 misidentification of the basal thermal state, as compared to direct observations (Table 1). The first area is NEEM, which is not surprising given the change in its assessed basal thermal state. The second area is the Prudhoe Lobe of the GrIS in far northwestern Greenland where Palmer et al. (2013) identified two subglacial lakes from radar sounding. While GBaTSv2 at the lake locations is uncertain, most of the rest of this lobe is likely frozen.

The assigned likely basal thermal state of 46% of the GrIS changed between GBaTSv1 and GBaTSv2; for < 6% of the GrIS, the assigned state changed from likely frozen to likely thawed or vice versa. GBaTSv2 identifies more of the bed to be likely frozen (+16%) and less to be likely thawed (–11%) than GBaTSv1. At first glance, this is surprising, because only 3-D thermomechanical models are used to identify a frozen bed in GBaTSv2. However, the loss of the discontinued method (onset of surface undulations) decreases the likelihood of a thawed bed identification, and the new method employed (basal water from airborne radar sounding) is inherently sparser in its more robust identification of a likely thawed bed (Fig. 4).

## 4 Discussion

A comparison of Fig. 3 in this study to Fig. 4 from M16 suggests that changes in bed topography influence thermomechanical model agreement on basal thermal state, as the pattern of agreement appears more focused in some regions, particularly along major outlet glaciers. Fig. 9a,b shows this difference in agreement in basal thermal state between the SeaRISE and ISMIP6 thermomechanical models. While we observe a possible relation between change in ice thickness and basal thermal state in the vicinity of several outlet glaciers, particularly in southern Greenland, the pattern is more nuanced across most of the ice-sheet interior. There is a noticeable divergence between northern and southern Greenland at around ~73ºN (Fig. 9b). This difference is not attributable to a new geothermal flux field, because most models from both ensembles use the older geothermal flux field derived from seismic data of Shapiro and Ritzwoller (2004), rather than a more recent field derived from aeromagnetic data (Martos et al., 2018) or machine learning (Colgan et al., 2022). Most ISMIP6 models used the BedMachine v3 bed topography, which on average results in thicker ice than the various bed topographies used by SeaRISE. A local increase in reference ice thickness between SeaRISE and ISMIP6 (Fig. 9a) would presumably tend to increase agreement where the bed is thawed, as the pressure-melting point at the bed will decrease. However, these changes are poorly correlated (linear correlation coefficient $r = 0.08$; Fig. 9c).

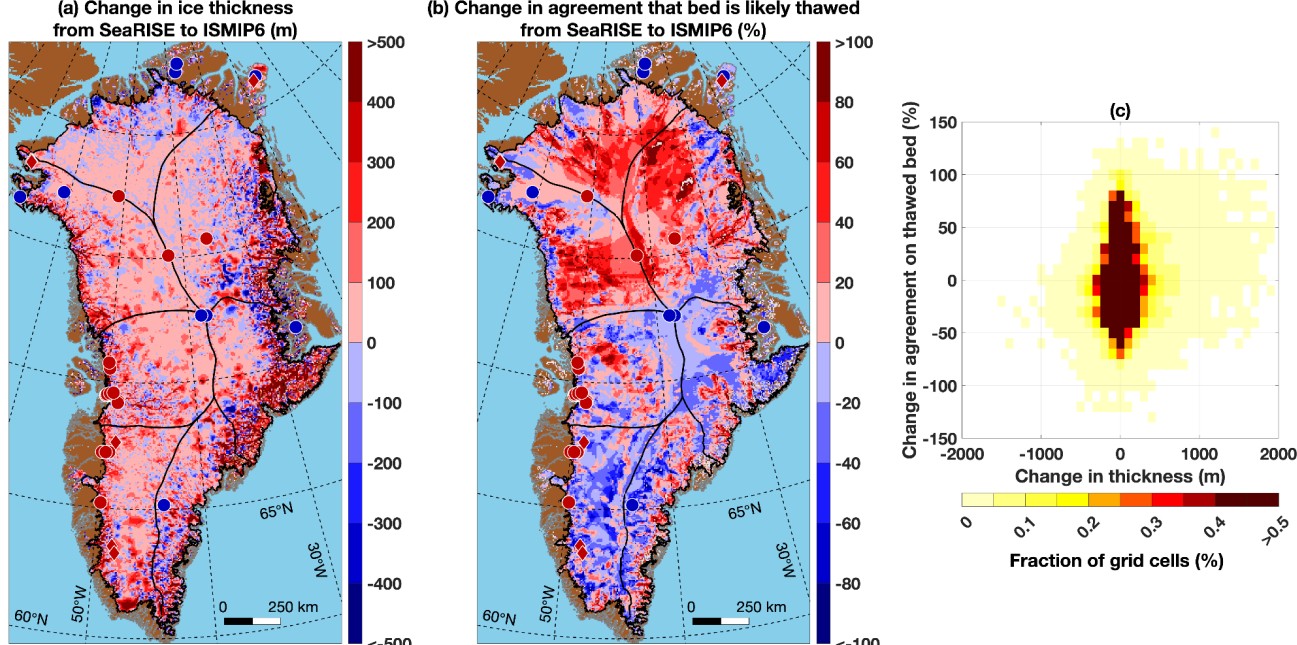

**Figure 9:** (a) Change in ice thickness between the most commonly used syntheses for SeaRISE (Bamber et al. (2001) with modifications) and ISMIP6 (Morlighem et al., 2017) on the 5-km grid used in this study. (b) Change in agreement in modeled basal thermal state from SeaRISE (Fig. 4 of M16) to ISMIP6 (Fig. 3), where a positive (negative) difference indicates greater agreement that the bed is thawed (frozen). Values greater than ±100% indicate that model agreement in the basal thermal state changed significantly. (c) Histogram of change in ice thickness vs. change in model agreement of a likely thawed bed.

A more likely explanation for the change in modeled basal thermal state is a mean change in the spin-up surface mass balance (SMB) field between the two ensembles, i.e., higher SMB in southern Greenland and lower SMB in northern Greenland for the ISMIP6 ensemble, along with possible changes in modeled surface paleotemperatures. Higher snowfall rates in the dry snow zone over multiple millennia lead to increased downward vertical advection and an overall colder ice column, which increases the likelihood of a frozen bed, and vice versa for lower snowfall rates. Unfortunately, the spin-up SMB fields used in SeaRISE and ISMIP6 models are more varied, so a simple comparison as in Fig. 9 for ice thickness cannot be generated easily to verify this hypothesis. Therefore, the root cause of the change in modeled basal thermal state remains not yet well understood.

While GBaTSv2 continues to be reported on a 5-km grid, it is increasingly clear that the basal thermal state can vary at scales finer than that (e.g., Chu et al., 2018). Further, englacial thermal structure can be quite variable at finer scales than 5 km (e.g., Lüthi et al., 2002, Harrington et al., 2015; Maier et al., 2019). Colgan et al. (2021) recently highlighted the role of bed topography in influencing geothermal flux at kilometer scales, a likely primary control on basal thermal state. However, this influence may be less important where there is negligible ice advection and basal temperature gradients are dominated by heat

diffusion (Willcocks et al., 2021). The sum of these studies suggests that finer-resolution geophysical methods and models are required to further specify the nature of Greenland's basal thermal state. This need could potentially be partly addressed by more intensive borehole investigations of regions where the basal thermal state is in question, especially in the deep interior of the GrIS and perhaps along existing flight lines where interpretations of airborne radar-sounding data disagree. Following the conclusion of OIB (MacGregor et al., 2021a), an opportunity now exists for an updated and more complete synthesis of basal water identifications from the data that mission collected, following existing methods (e.g., Jordan et al., 2018; Chu et al., 2018).

We next consider the impact of the GBaTSv2 revision upon several existing studies that consider either the ice sheet's basal thermal state or used GBaTSv1 explicitly. Poinar et al. (2015) concluded that surface-melt-induced acceleration of ice flow is unlikely to propagate inland significantly within the SW basin of the GrIS, partly because they modeled that the bed is mostly thawed farther inland there. Although GBaTSv2 indicates decreased confidence in a thawed bed beneath the upper reaches of SW basin of the GrIS, as compared to GBaTSv1, the extent of this change does not appear to impact the conclusions of Poinar et al. (2015). Maier et al. (2021) found that inferences of Greenland's regional basal rheology were insensitive to which portion of each ice-drainage basin was identified as likely thawed or uncertain in GBaTSv1. For most basins, the combined region identified as likely thawed or uncertain in GBaTSv2 is similar in extent to GBaTSv1's likely thawed region, so we expect that Maier et al. (2021)'s conclusions regarding Greenland's regional basal rheology are not substantially affected by this revision. A similar assessment applies to Karlsson et al. (2021)'s estimate of GrIS basal meltwater production. They used GBaTSv1 to constrain the geothermal and frictional contributions to basal melt. Because frictional heating is concentrated within typically thawed outlet-glacier systems whose likely basal thermal state is mostly unchanged from GBaTSv1 to GBaTSv2, the frictional contribution to basal melt is unlikely to change significantly except for some eastern outlet glaciers south of ~75ºN, where it may decrease. For the geothermal contribution to basal melt, GBaTSv2 suggests that southern drainage basins may produce less basal melt, while northern drainage basins may produce more. The net effect of the GBaTSv2 revision is likely that the total estimated GrIS basal melt is less than that reported by Karlsson et al. (2021), but it is unlikely that this change exceeds the ~20% relative uncertainty in their estimate.

It is unlikely that the basal thermal state of the GrIS will significantly affect its evolution over this century, which is the period considered by ISMIP6 (Goelzer et al., 2020). However, if anthropogenic climate forcing persists beyond this century and continues for a substantial portion of this millennium, then GrIS retreat will likely be substantial and its present basal thermal state will have a progressively greater influence upon the nature of this retreat, because submarine melting will very likely outpace thermal diffusion at the bed (Aschwanden et al., 2019). Because observed GrIS mass loss presently tracks at the upper end of the range projected by the ISMIP6 ensemble and the socioeconomic impacts of rising sea levels are vast (Aschwanden et al., 2022), it remains essential to produce ice-sheet-wide assessments of basal boundary conditions, such as GBaTSv2, which can help validate model simulations of the present state of the GrIS. Such efforts will also ultimately help increase confidence in model projections of future sea-level rise beyond this century.

## 5 Data availability

The core result of this study, version 2 of the likely basal thermal state of the Greenland Ice Sheet (GBaTSv2; Fig. 8b), is available at https://doi.org/10.5281/zenodo.5714527 (MacGregor et al., 2021b) and will also be later made available through the National Snow and Ice Data Center, where further dataset-specific documentation will be provided. This dataset also includes the syntheses of other freely available datasets shown in Figs. 3, 4, 5d, 6c and 7b/c/d.

## 6 Conclusions

We have developed and presented the second version of the likely basal thermal state of the Greenland Ice Sheet (GBaTSv2). This second estimate is broadly similar to the first, although there is substantial regional variability therein and a greater tendency toward a likely frozen basal thermal state. The large-scale similarity is likely due to the applied methods being mostly replicated from the first version, despite underlying updates to associated datasets and the discontinuation of one method. This new synthesis suggests that the bed of the GrIS is roughly equal parts thawed (33%), frozen (40%) or too uncertain to specify (28%). Although the use of an improved bed topography beneath the GrIS within 3-D thermomechanical models does not appear to be related to greater agreement in basal temperature within those models, we do observe more spatially focused patterns of likely thawed bed within outlet-glacier systems that have been better mapped since GBaTSv1. The effect of these revisions upon existing studies that used GBaTSv1 is likely to be modest, but the influence of the basal thermal state upon ice flow is likely to increase if anthropogenic climate forcings persists beyond this century. Absent future investigations to directly measure basal temperature in new boreholes, to more extensively identify basal water from remote sensing and to map likely pathways for that basal water, the suite of methods we employed may be approaching a natural limit in its ability to resolve the basal thermal state. Future syntheses should consider new, finely resolved yet ice-sheet-wide observations, which will most likely come from further campaigns or advances in airborne or satellite remote sensing.

**Author contribution.** JAM initiated this study, led the analysis and drafted the manuscript. WTC, NBK and SMJN provided new and existing datasets, contextualized them, aided interpretation and edited the manuscript. WC, DF, MAF and MS interpreted the analysis and edited the manuscript.

**Code availability.** The MATLAB script used to perform the analysis and generate the figures in this manuscript is available at https://doi.org/10.5281/zenodo.5714527. Most of the analysis was performed using functions built-in to MATLAB R2022a with its Mapping and Image Processing toolboxes.

**Competing interests.** Joseph A. MacGregor is a member of the editorial board of this journal and Nanna B. Karlsson is a Co-Editor-in-Chief of this journal.

**Acknowledgements.** We thank the NASA/GSFC Internal Scientist Funding Model for supporting the generation of this dataset. We thank D. Dahl-Jensen for valuable discussions concerning NEEM and EastGRIP, E. Simon, Y. Choi, N. Schlegel, V. Lee and H. Seroussi for valuable discussions concerning ISMIP6 models, and A. Fitzgerrell for assistance with NetCDF design. With respect to our of use of ISMIP6 model outputs: 1. We thank the Climate and Cryosphere (CliC) project for supporting this project and the ice sheet modelers who participated in it; 2. We acknowledge the World Climate Research Programme and its Working Group on Coupled Modelling for coordinating and promoting CMIP5 and CMIP6; 3. We thank the climate modeling groups for producing and distributing their model output; the Earth System Grid Federation (ESGF) for archiving the CMIP data and providing access, the University at Buffalo for ISMIP6 data distribution, and the funding agencies who support CMIP5, CMIP6 and ESGF; and 4. We thank the ISMIP6 steering committee, model selection and dataset preparation groups for defining ISMIP6. Finally, we thank the Editor A. Robinson, referee A. Aschwanden and an anonymous referee for constructive comments that improved the manuscript.

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
