# Peer review of "GBaTSv2: A revised synthesis of the likely basal thermal state of the Greenland Ice Sheet"

_The Cryosphere, 2022_

## Author Comment (AC1)

**Response to reviews of "GBaTSv2: A revised synthesis of the likely basal thermal state of the Greenland Ice Sheet"**

Joseph A. MacGregor et al.
8 June 2022

We thank the Editor A. Robinson, referee A. Aschwanden and an anonymous referee for their constructive and quite positive comments on this manuscript. We've addressed all of them below, with first the comment in italicized blue and then our response in black. We have also made several minor editorial adjustments, included some additional statistics on the change from v1 to v2, and expanded the Discussion to better reflect what was promised in the Introduction section (discussing the implications of GBaTSv2 on interpretations of ice flow). All changes were recorded via tracked changes. Finally, we also revised our calculations slightly to exclude peripheral ice masses more reliably from our ice-sheet statistics. We hope that the revised manuscript is satisfactory for publication.

**Response to A. Aschwanden**

*With >100 citations on Scopus, the manuscript describing GBaTSv1 has been a huge success, demonstrating a clear community need. I applaud the authors for their (most likely tedious) effort of updating the product and making it again available to the community; I have no doubt that GBaTSv2 will become equally successful as its predecessor.*

*The methodology closely follows v1, and any deviations are carefully motivated. This facilitates comparison to v1 and also makes reviewing this manuscript relatively easy. The author's writing is, as usual, impeccable and shows attention to detail, making the manuscript a breeze to read.*

*I downloaded the data set and imported it into QGIS, everything worked without a hitch.*

We thank the referee for their positive assessment of the manuscript, recognition of its broader value and clear understanding of our methodology. We're glad the dataset works well in QGIS as this was an issue with v1 and we spent considerable effort this time around to both improve our NetCDF formatting and ensure GIS compatibility, with the substantial help of NSIDC.

*Subscripts that are not variables should be in \mathrm{}, e.g. $\rho_{\mathrm{ice}}$.*

Agreed. We've addressed this issue for spelled out subscripts but also abbreviated several of the subscripts as appropriate (e.g., "ice" is now "i", "bed" is now "b").

*Fig 2: I wonder if the color map could be improved to better visualize the difference between ice at and ice below the pressure melting point (i.e. the threshold chosen in this study). Maybe only use red of at the PMP, and other colors/shades for below?*

We prefer the deep red for the pressure-corrected basal temperature range between 0 and –1ºC, as we believe it is readily distinguished from lower temperatures / lighter reds. We disfavor single-color palettes as they tend to be hard to distinguish. We've added an annotation to the color bar to make clearer that this deep red means thawed using the standard temperature threshold.

*Fig 6: what is $\tilde u_s$?*

Uncertainty in surface speed, now clarified in text.

> *Fig 7. May I suggest to use line colors that are color-blind friendly? See https://colorbrewer2.org for inspiration.*

Agreed and fixed. This was an oversight due to repurposing figure generation code from v1.

**Response to anonymous referee**

> This manuscript updates the previously published basal thermal state of the Greenland Ice Sheet (GrIS), GBaTSv1, from MacGregor et. (2016). I welcome this updated product with updated methodology and use of more recent modeled and observed products. I have no doubt it will be of great use to the scientific community.
>
> I found the manuscript to be very well written and organized with enough information to make it approachable to general readers. I appreciated the authors took the time to pinpoint approaches that were both similar and different compared to the previous product, easing the comparison between both products. Also, I was delighted about the effort spent on the figures which are easy to follow and understand. I wish all submitted manuscript would be of this quality!
>
> I was able to download the data very easily.

We thank the referee for their positive assessment of the MS and appreciation of its content, structure and style. We're also glad the dataset was easily downloadable as this was our intention. We will also be submitting this dataset as a candidate to be archived at NSIDC.

> In your paper, you list the names and location of some deep borehole in table 1. You locate NorthGRIP and Summit in Fig8. I would find it useful to locate NEEM, DYE-3, and Prudhoe lobe on the same map as well. Maybe it would be suitable to locate them all in Fig.1 especially since you explain the change of state of the NEEM borehole from likely frozen to likely thawed in Sec. 2.1.

Agreed. We had wavered on making this adjustment previously because most of the sites were shown in the 2016 paper. However, we've now updated Fig. 1 with labels for all sites mentioned in the text and removed them from Fig. 8a for simplicity.

> Throughout the text you make several references to Cuffey and Paterson 2010 (e.g., p2, line2). Since this is a book that is over 650 pages long, I would suggest you adding the chapter and/or the page number/range of the book as you refer to it.

Agreed and now addressed throughout the MS except for the first citation in the Introduction section, where the citation is intended as general motivation for the problem of classifying the basal thermal state.

> This is me being curious here: did you find any correlation in diagnosing the basal thermal state between ISMIP6 models using the SIA or hybrid SIA (if any) along with surface velocity from Joughin et al. (2016, 2017) and your method of minimum basal slip ratio? (I know it is more complicated than just using SIA in an ice sheet model, but still, I'm curious.)

That is a good question that we partly considered in the 2016 paper but did not address in detail here. We didn't directly compare SIA ISMIP6 models to the minimum basal slip ratio method, as we seek to avoid a direct comparison with individual ice-sheet models, which each have their strengths and weaknesses. Perhaps the closest we get is between individual panels in Fig. 2 and Fig. 7a, which outlines the thawed extent from the minimum basal slip ratio method. In this scenario, the models with the greatest thawed extent (e.g., UAF/PISM, VUW/PISM, NCAR/CISM, LSCE/GRISLI2) are more similar to the minimum basal slip ratio method.

> P8, line 179: Do you mean "Uncertainty" instead of "Uncertainly"?

We did indeed. Thanks for spotting this typo.